# The Wound-Healing Activity of PEDOT-PSS in Animals

**DOI:** 10.3390/ijms241612539

**Published:** 2023-08-08

**Authors:** Yun-Lung Chung, Pei-Yu Chou, Ming-Jyh Sheu

**Affiliations:** 1School of Pharmacy, College of Pharmacy, China Medical University, Taichung 406040, Taiwan; p730912@hotmail.com; 2Department of Nursing, National Chi Nan University, Nantou 54561, Taiwan; peiyu67@gmail.com

**Keywords:** wound-healing, angiogenesis, VEGF, TGF-β1

## Abstract

This study evaluated the wound-healing activity of a polymer, Poly(3,4-ethylenedioxythiophene):poly-(styrene sulfonate) (PEDOT: PSS), and determined its mechanism based on angiogenic activity in a full-thickness excision wound model in Spraque Dawley (SD) rats. Administering PEDOT: PSS (1.6) 1.5 ppm at a dose of 50 mg/kg/day significantly improved wound healing in the SD rats on the eleventh day after the incision was created. PEDOT: PSS-treated animals presented no anti-inflammatory skin effects; however, there was an increase in angiogenic behavior. VEGF was found to be significantly elevated in the PEDOT: PSS-treated groups seven days post-incision. However, only a higher concentration of PEDOT: PSS increased TGF-β1 expression within the same time frame. Our results showed that PEDOT: PSS enhances wound healing activity, mainly in terms of its angiogenic effects. In this paper, we describe the highly conductive macromolecular material PEDOT: PSS, which demonstrated accelerated wound-healing activity in the animal incision model. The results will further provide information regarding the application of PEDOT: PSS as a dressing for medical use.

## 1. Introduction

Wound healing is a complex process, and the injured tissue must undergo complicated physiological processes to restore normal functions. Although the wound-healing process is not yet fully understood, it is known to involve the following tissue remodeling phases and their associated physiological actions: a vascular and inflammatory phase (hemostasis and inflammation), a tissue repair phase (fibroproliferation and cellular proliferation), and a tissue remodeling phase. Of these processes, inflammation in particular is a crucial stage of wound healing [1,2]. Inflammatory symptoms are prevalent at all stages of wound healing. In the early stages, neutrophils will quickly gather in the wound areas. The primary function of neutrophils is to kill and remove microorganisms, such as bacteria and fungi [3]. Afterward, monocytes begin to aggregate and transform into macrophages. Macrophages rapidly increase in number and, in turn, replace neutrophils, becoming the dominant leukocytes by the third day after wound formation [2,4]. Macrophages play a significant role in wound healing. The main reason is that this process produces many cytokines, chemokines, and growth factors [5]. The growth factors involved in wound healing mainly include vascular endothelial growth factor (VEGF), TGF-β1, and epidermal growth factor (EGF) [6]. These growth factors initiate the formation of granulation tissue, which gradually proliferates microvascular, fibrous connective tissue to allow the wound to heal quickly. In addition, many inflammatory cells are produced. This gradual appearance of tissue covers the entire microvascular system, and the tissue will gradually turn bright red. In medicine, this is called granulation tissue, and it is known to promote the synthesis of new collagen, proteoglycan, and insoluble fibronectin. This step plays an essential role in the dermal restoration process. The remodeling phase typically occurs within months to years of initial wound healing and involves the extracellular matrix and repair of the wound site at all levels. There is a need for further studies supporting the vital role fibronectin plays in the wound-healing process [7].

Although certain pharmacological agents expedite wound healing, certain medications can cause additional wounds by damaging the integrity of the skin. Therefore, it is important to find an option that will help with wound healing and also prevent skin injury [8]. Poly(3,4-ethylenedioxythiophene):poly-(styrene sulfonate) (PEDOT: PSS) is a conductive macromolecular material, in which PEDOT is a conjugated polymer carrying positive charges, whereas the sulfonyl groups of PSS offer negative charges. PEDOT: PSS has been applied in many biomedical devices [7] and bio-conductive applications due to its high electrical conductivity, water dispersibility, and, most importantly, its favorable cytocompatibility [9,10,11]. This study explores whether PEDOT-PSS, in a hydrogel formulation, can effectively promote wound healing and aims to understand its molecular mechanism.

Here, we demonstrated that PEDOT: PSS (1.6) in an aqueous solution shows no cytotoxicity to macrophage cells (RAW 264.7) (Figure 1). Our results showed that the polymer PEDOT: PSS (hydrogel form) greatly enhanced wound-healing activity 11 days post-incision in the animal model (Figure 2). The molecular mechanisms by which PEDOT: PSS hydrogel formulas enhance wound healing were also examined. VEGF has been shown to be upregulated after treatment with PEDOT-PSS in animals (Figure 3). TGF-β1 also showed a slight increase in the PEDOT: PSS-treated animals (Figure 4). PEDOT: PSS (aqueous form) significantly increased wound healing activity in vitro (Figure 5). Our work emphasizes that the PEDOT: PSS hydrogel formula is a material that could potentially accelerate wound healing in the human body with extremely low cytotoxicity.

## 2. Results

### 2.1. PEDOT-PSS (1.6) Is Non-Toxic to Human Macrophages

In this experiment, human macrophages (RAW 264.7) were treated with different concentrations of PEDOT-PSS (1.6), PEDOT-PSS (2.5), and PEDOT-PSS (5.0) to determine their effects on cell viability. It was found that, compared with other types of PEDOT-PSS, PEDOT-PSS (1.6) did not cause cell cytotoxicity (Figure 1). As PEDOT: PSS (1.6) is non-toxic to the human macrophage, PEDOT-PSS (1.6) was chosen for our studies in vitro and in vivo.

### 2.2. Pathological Incidences of PEDOT: PSS (1.6) on Wound Healing in the Full-Thickness-Excision Wound Model in Rats

PEDOT: PSS-treated animals presented with inflammatory skin effects; however, there was also an increase in angiogenic behavior in the wound area at day 7 post-incision (Table 1). PEDOT: PSS-treated animals presented the least inflammatory effects on the animal’s skin at day 14 post-incision (Table 1) and also exhibited prominent angiogenic behavior, granulation, and re-epithelialization of the wound (Table 1).

### 2.3. PEDOT-PSS (1.6) Significantly Accelerates Animal Wound-Healing Behavior

The results of the animal studies showed that PEDOT-PSS (1.6) 1.5 ppm significantly accelerated wound healing in the SD rats eleven days post-incision compared to the control group (Figure 2), especially after 11 days of incision. (Figure 2, Table 2). * *p* < 0.05, compared to the control group.

### 2.4. PEDOT-PSS (1.6) Significantly Increased the Expression of Vascular Endothelial Growth Factor (VEGF) during the Animal Wound-Healing Process

In animal experiments, PEDOT-PSS (1.6) 1.5 ppm markedly increased the VEGF and TGF-β expression in the endothelial cells of the granulation tissue. The statistical results showed that, when the concentration of PEDOT-PSS (1.6) was 1.5 ppm for wound treatment, the staining intensity of VEGF was significantly different at *p* < 0.05 compared to the control group; its staining frequency was significantly different at *p* < 0.05 compared with the control group at seven days, but not fourteen days, after the wound incision (according to Student’s *t*-test) (Figure 3, Table 3).

### 2.5. PEDOT-PSS (1.6) in Higher Concentrations had Significant Effects on the Expression of TGF-β1 during the Animal Wound Healing Process

In animal experiments, PEDOT-PSS (1.6) 1.5 pm increased the TGF-β1 expression in the endothelial cells of the granulation tissue. The statistical results showed that, when PEDOT-PSS (1.6) 1.5 ppm was given to treat the wounds, the staining frequency was markedly increased compared to that of the control group (Figure 4, Table 4).

### 2.6. PEDOT-PSS (1.6) Significantly Promotes Wound Healing In Vitro

Our results showed that PEDOT-PSS (1.6) 1.5 ppm significantly enhances the migration activity of macrophages (macrophage; RAW264.7) via a wound-healing assay. When PEDOT-PSS (1.6) 1.5 ppm was given, macrophages (macrophage; RAW264.7) demonstrated significant wound healing/closure (1.6) in the 1.5 ppm treated group compared to that of the control group after 24 h of exposure (Figure 5). * *p* < 0.05 (Student’s *t*-test)

### 2.7. The Physical Properties of PEDOT: PSS

The PSS: PEDOT molar ratios for PEDOT:PSS (1.6), PEDOT: PSS (2.5), and PEDOT: PSS (5.0) are 1.6, 1:2.5, and 1:5.0, respectively. In the design of PEDOT: PSS, the greater the amount of PSS, the poorer the conductivity of the material (Table 5). This is because PSS is non-conductive. Meanwhile, PEDOT: PSS shows the greatest conductivity. The reason that PEDOT: PSS is not available for our study is that PEDOT: PSS (1.0) is fairly unstable in water solution and must be stored at a lower temperature. Our results also showed that PEDOT: PSS showed more intensity compared to PEDOT: PSS (2.5) and PEDOT: PSS (5.0) (Figure 6).

## 3. Discussion

PEDOT: PSS is a “one-dimensional” linear polymer with conformationally mobile chains. This characteristic demonstrates its superior adhesion and penetration. PEDOT: PSS presents with separated positive and negative charges that consist of linear flexible molecules, which appear to be preferable for biological applications [12]. Importantly, PEDOT: PSS is devoid of toxicity and shows no carcinogenic potential. Therefore, PEDOT-PSS (1.6) is expected to be developed into biological products. According to our results, PEDOT-PSS (1.6) demonstrated the lowest level of cytotoxicity in human macrophage cells (RAW 264.7). The cells were treated with PEDOT-PSS (1.6) at different concentrations, even up to 1500 ppm, for 24 h, and the cells did not exhibit cytotoxicity (Figure 1). However, further animal toxicity tests (including 28 days, 90 days, or even genotoxicity tests) must be conducted. Our results found that PEDOT-PSS (1.6) did not cause tissue inflammation after 14 days of administration to animals (Table 1). In the animal study, our results revealed that PEDOT-PSS (1.6) 1.5 ppm significantly improved the wound-healing activity of the animals compared to the control group, especially after 11 days post-incision (Figure 2, Table 2). Moreover, the animals treated with PEDOT-PSS (1.6) 1.5 ppm showed a significantly increased VEGF expression (Figure 3, Table 3). Additionally, TGF-β1 showed a slight increase in the PEDOT: PSS-treated animals (Figure 4). Our results found that PEDOT-PSS (1.6) at a concentration of 1.5 ppm significantly increases wound healing/closure compared to the control group in vitro (Figure 5).

It is recognized that macrophages play important roles in regulating wound-healing activity, exhibiting remarkable plasticity and an evolving phenotype, from a pro-inflammatory to a pro-healing phenotype, to guarantee an appropriate healing process [13]. The dysregulation of macrophages correlates with the impaired healing of diabetic wounds. This helps us recognize macrophages as a potential target to enhance wound-healing activity [14]. Macrophages have the most critical influence on wound healing because they produce many cytokines, chemokines, and growth factors [3]. VEGF, TGF-β1, and EGF are involved in wound-healing activity [6,15,16]. These growth factors contribute to granulation tissue formation. Immediately after the skin is injured, the nearby tissue gradually proliferates microvessels, fibrous connective tissue, and a large number of inflammatory cells. After two days, the granulation tissue forms. This will further the development of collagen, proteoglycan, and fibronectin. Then, angiogenesis influences wound healing by producing neo-vessels and reconstructing the dermal–epidermal junction using myofibroblasts. Our study indicated that treatment with PEDOT-PSS (1.6) 1.5 ppm clearly demonstrated the overexpression of VEGF seven days post-incision (Figure 2).

VEGF is usually present in low concentrations in epidermal keratinocytes but increases during wound healing [17]. The main reason for this is hypoxia, which is one of the reasons for the increase in VEGF during wound healing, so the increased concentration of VEGF means that it stimulates wound healing [16]. Importantly, our results showed that after treatment with PEDOT-PSS (1.6) in animals, the expression of VEGF was significantly increased by immunohistochemical staining on the seventh day after wound formation (Figure 3). The expression of VEGF increased significantly on the seventh day, which means that the tissue underwent angiogenesis, and this action supports the process of tissue wound healing [18]. VEGF represents the primary regulator of angiogenesis, and its existence is related to this process. During wound healing, supplying sufficient nutrients to the wound site is vital for angiogenesis [18]. These results concluded that PEDOT: PSS can indeed significantly increase the expression of VEGF, promoting wound-healing activity (Figure 3, Table 3).

TGF-β1 plays a significant role in wound-healing activity by influencing several phases, including hemostasis, inflammation, proliferation, and remodeling [19]. At the beginning of the wound-healing process, TGFβ1 stimulates inflammation in the wounds [18]. TGF-β1 increases the proliferation of granulation tissue during wound healing and other relevant processes [20]. TGF-β1 promotes the proliferation of fibroblasts during wound healing, which then activates fibroblasts to differentiate into myofibroblasts cells (myofibroblasts) [21] and promotes collagen synthesis [22]. TGF-β1 participates in several principal effects on the wound-healing process, including the induction of fibroblast proliferation [6,21,22]. Certain agents that trigger wound healing have been correlated with high TGF-β1 expression accompanied by fibroblast proliferation in vivo [23]. The results showed that PEDOT: PSS treatment only slightly enhanced TGF-β1 protein expression in granulation tissue (Figure 4). This could explain why TGF-β1 may present a higher number in the early stage of the incision made in the animal model. 

Our results showed that PEDOT-PSS (1.6) did not cause tissue inflammation after 14 days of animal treatment. This means that this polymer substance does not cause external inflammation of the tissue and may even inhibit the inflammation of the tissue. However, this needs to be determined in a future study. Therefore, there is no correlation between the inflammation of the tissue during the healing process and the treatment with PEDOT-PSS (1.6). NF-κB1 and tumor necrosis factor (TNF-α) are two important biomarkers of inflammation [24]. Studies showed that apparent inflammation of the incision is associated with TNF-α and NF-κB elevation [6]. The reasoning behind this is that unnecessary inflammation brings about delayed wound healing, resulting in a chronic wound [6]. Therefore, our results found that this polymer does not affect tissue inflammation and may play a positive role in wound healing. As to whether PEDOT-PSS can inhibit NF-κB1 and TNF-α, further verification is needed.

This research first described that the polymer PEDOT: PSS effectively accelerates wound-healing activity in vitro and in vivo. Further studies should investigate its suitability for co-administration with the active agent for wound healing in clinics. Particular focus should be given to chronic wounds, such as those experienced by diabetes mellitus patients.

## 4. Materials and Methods

### 4.1. Materials

Macrophages (macrophage; RAW264.7) were purchased from Bioresource Collection and Research Center (BCRC; Hsinchu, Taiwan). Primary antibodies, anti-VEGF (product code number: ab32152), and anti-TGF-β1 (product code number: ab215715) were purchased from Abcam. Penicillin and streptomycin were obtained from Lonza (P/S; Walkersville, MD, USA). Fetal bovine serum (FBS) was bought from GIBCO (Gaithersburg, MD, USA). Goat anti-mouse IgG and goat anti-rabbit IgG secondary antibodies were obtained from Jackson ImmunoResearch Laboratories (West Grove, PA, USA). PEDOT: PSS aqueous solution (with a 1:1.6, 1:2.5, and 1:5.0 ratio of PEDOT to PSS, respectively) was used for the in vitro study (provided by DAILY POLYMER Corporation, Kaohsiung city, Taiwan) [25,26]. The hydrogel was designed for the wound-healing assay in vivo. The polymer, PEDOT: PSS (1.6), PEDOT: PSS (2.5), and PEDOT:PSS (5.0) samples are aqueous dispersions, with the main components being poly(3,4-ethylenedioxythiophene) (PEDOT) and poly(styrenesulfonate) (PSS). The synthesis involves the polymerization of EDOT monomers with different ratios of PSS:PEDOT. The PSS: PEDOT molar ratios for PEDOT:PSS (1.6), PEDOT: PSS (2.5), and PEDOT:PSS (5.0) are 1:1.6, 1:2.5, and 1:5, respectively. Different ratios result in different structures and variations in conductivity, aiming to investigate cytotoxicity and compatibility issues. Moreover, to transform the solution-state PEDOT: PSS into a hydrogel for better adhesion during animal experiments, an aqueous thickening agent (provided by the First Cosmetics Works Limited, Taoyuan city, Taiwan) is added at a proportion of 1% of the total solution. This transformation helps effectively explore the impact of wound healing in animal studies, especially with regard to easy application of hydrogen to the animals’ wounds.

### 4.2. Cell Culture

Cells were cultured in 96-well Dulbecco’s modified Eagle’s medium (DMEM) medium supplemented with penicillin, streptomycin, and 10% fetal bovine serum at 37 °C with 5% carbon dioxide before in vitro cell assays. The experimental medium was 90% DMEM plus four mM L-glutamine (adjusted to contain 1.5 g/L sodium bicarbonate and 4.5 g/L glucose + 10% fetal bovine serum).

### 4.3. Cell Viability Analysis

Human macrophage cell lines (RAW 264.7) were cultured in Dulbecco’s modified minimal essential medium (DMEM) in an incubator at 37 °C. When the cells reached a steady state, the cells (3 × 10^4^ cells) were evenly distributed on the multi-well plate and then returned to the incubator to continue growing overnight. After that, PEDOT-PSS (1.6, 2.5, and 5.0) at various concentrations (2.9, 5.9, 11.7, 23.4, 46.9, 93.8, 187.5, 375, 750, and 1500 ppm) were added to each well, and three replicates were performed. After 24 **h**, PBS was added for washing, and a further 10 μL of the MTT was add into each well and incubated at 37 °C for 2 h. Finally, the purple formazan crystals produced by the viable cells were measured at the absorbance length (590 nm) using an ELISA reader (Bio-Tek, Winooski, VT, USA).

### 4.4. Wound-Healing Assay

Macrophages (RAW 264.7; 2.5 × 10^4^ cells/well) were seeded into 24-well dish plates. After 24 h, when the cells were confluent, a 200 μL micropipette tip was used to create a cell-free gap, and then PEDOT-PSS (1.6) was administered to the culture medium. At 0 and 24 h after cell treatment, the cell-free space was photographed, and the area of the cell-free space was measured via ImageJ 1.44 software analysis (NIH, Bethesda, MD, USA). Migration inhibition was expressed as a percentage of wound closure based on calculation of the cell-free space for each group compared to the initial time (0 h).

### 4.5. Animal Experiment

Sprague Dawley (SD) rats were anesthetized with Zoletil (Zoletil 50) at a concentration of 50 mg/mL at a dose of 0.5 cc/kg (intramuscular (IM) injection). Experimental animals were fed in the animal room of China Medical University. The light cycle of the animal room was adjusted automatically, following a cycle of 12 h day and 12 h night. The temperature in the breeding room was approximately 25 °C, and there was no fasting, water deprivation, or movement restriction during the experiment. The experiment’s procedure is detailed as follows: SD rats (300–400 g) first stayed in the animal room for a week in individual cages. After that, the hair on the back surface was shaved (a wound placed in this area cannot be accessed by the rat, preventing self-licking) and then sterilized with 70% alcohol, and the injury was made on the back under anesthesia according to the weight of the animal (wound area: 20 × 20 mm). The SD rats were divided into four groups (*n* = 5). Treatment was administered to each animal every day, including the control group (Vaseline ointment; 100 mg/kg) and the PEDOT-PSS (1.6) experimental group, at concentrations of 150 ppm, 15 ppm, and 1.5 ppm for treatment (50 mg per application/day). The drug was applied once a day. Wound photographs were taken on days 3, 7, 11, and 14 after surgery. Only on the 7th and 14th day after the wound was created was the tissue removed (for each group of cross-sections of granulation tissues, a Bistoury surgical knife was used to extract a tissue range of about 20 × 20 mm, and the depth was about 2 mm for the primary extraction. The sampling range included the epidermis, subcutaneous, dermis, and meat membrane). Afterward, H&E staining was performed, as well as IHC immunohistostaining analysis. The lower extremity tissue was removed, and then H&E staining was performed to investigate the inflammation. Immunohistochemistry was then performed to determine the differences in the expression of vascular endothelial growth factor (VEGF), TGF-β, and other proteins. This study was approved by the Institutional Animal Care and Use Committee (IACUC) (animal review number: 2022-463-1).

### 4.6. H&E Stain

The animal granulation tissue was removed and perfused with 4% paraformaldehyde for 30 min. The samples were fixed in 4% paraformaldehyde for 24 h, embedded in paraffin, and cut into 7 mm cross-sections. The animal tissue sections were deparaffinized and rehydrated via gradient elution of ethanol and cleared in xylene, followed by staining with hematoxylin–eosin Y. The slides were measured and photographed using an Olympus BX53 fluorescence microscope (Tokyo, Japan). The staining intensity was analyzed using ImageJ 1.44 software (National Institutes of Health, Bethesda, MD, USA).

### 4.7. Immunohistochemistry

Firstly, for the tissue dewaxing, xylene was used in 2 cylinders for 10 min each. This was followed by tissue backwater,(decreasing alcohol) 100% alcohol, 95% alcohol, 80% alcohol, 70% alcohol, and then water washing in each tank for 3 min. An antigen retrieval pot was used for antigen retrieval, which took approximately 10–20 min. This was followed by washing with TBST for 5 min. Next, the freezing step occurred; this involved using a hydrogen peroxide block for 10 min. Two cylinders of TBST were used; the cleaning time lasted 5 min per sample. When a protein block was used, this action took 10 min. Then, two cylinders of TBST were used; the cleaning time was 5 min for each sample. The tissue’s primary antibody was stained for about 30 min. Then, two cylinders of TBST were used. The cleaning time was 5 min per sample. Afterward, Primary Antibody Amplifier (Quanto) was applied for 10 min. Two cylinders of TBST were used for 5 min each. Then, HRP polymer (Quanto) was used for 10 min. Two TBST cylinders were used, and the cleaning time lasted 5 min for each sample. The tissue was stained with DAB Chromogen: Substrate = 1:30, and the coloring time was about 1–3 min. Then, the tissue was washed with tap water for 5 min. The tissues were stained with hematoxylin, and the staining time was 1–5 min. Then, the tissue was washed with tap water for 10 min. The water on the slide was shaken off, and the sample was dehydrated with (increasing alcohol) 80% alcohol, 95% alcohol, 100% alcohol, and 100% alcohol for each tank for 5 min. Next, the tissue was soaked in xylene in two cylinders for 5 min each. Finally, the tissue was sealed.

### 4.8. Statistical Analysis

The statistical analysis was performed using the SPSS Statistics software (vers. 19; IBM, Armonk, NY, USA). All the independent values are represented as means ± standard deviations (SD). In the multiple-group tests, a one-way analysis of variance (ANOVA) followed by a post hoc test (Dunnett’s test) was run to verify the statistical significance. A *p*-value < 0.05 was believed to be statistically significant. Student’s *t*-test was used to analyze the significance between two IHC and in vitro wound-closure intervention groups in the cell viability assay.

## Figures and Tables

**Figure 1 ijms-24-12539-f001:**
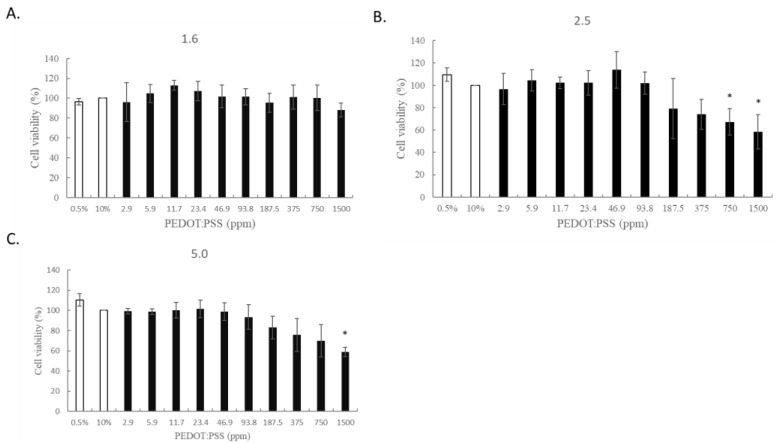
Cytotoxic effects of PEDOT: PSS on the human macrophage (RAW 264.7 (**A**) PEDOT-PSS (1.6) showed non-toxic effects in human macrophages (RAW 364.7). The cells were treated with different concentrations of PEDOT-PSS (1.6) (2.9, 5.9, 11.7, 23.4, 46.9, 93.8, 187.5, 375, 750, and 1500 ppm) for 24 h. These cells did not show apparent cell cytotoxicity. (**B**) PEDOT-PSS (2.5) and (**C**) PEDOT-PSS (5.0) at higher concentrations showed marked cell cytotoxicity. * *p* < 0.05, compared to the control group (treated with 10% FBS alone), according to a one-way analysis of variance (ANOVA).

**Figure 2 ijms-24-12539-f002:**
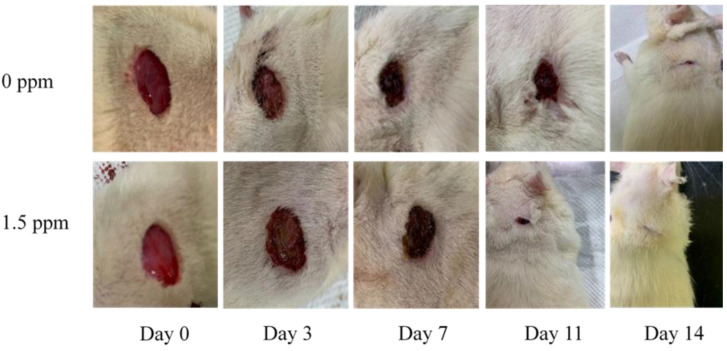
Effect of PEDOT: PSS (1.6) on the wound healing process in SD rats. SD rats (*n* = 5) were administered with PEDOT: PSS (1.6) 1.5 ppm. It was administered to the control group with Vaseline ointment (100 mg per application/kg/day) and the PEDOT-PSS (1.6) experimental group at concentrations of 1.5 ppm for treatment (50 mg per application/day). Wound areas of the animals significantly reduced at day 11 and day 14 after the full-thickness excision compared to the control group (Day 0) (Table 1). Wound photographs were taken on days 3, 7, 11, and 14 after the incision. *p* < 0.05, compared to the control group.

**Figure 3 ijms-24-12539-f003:**
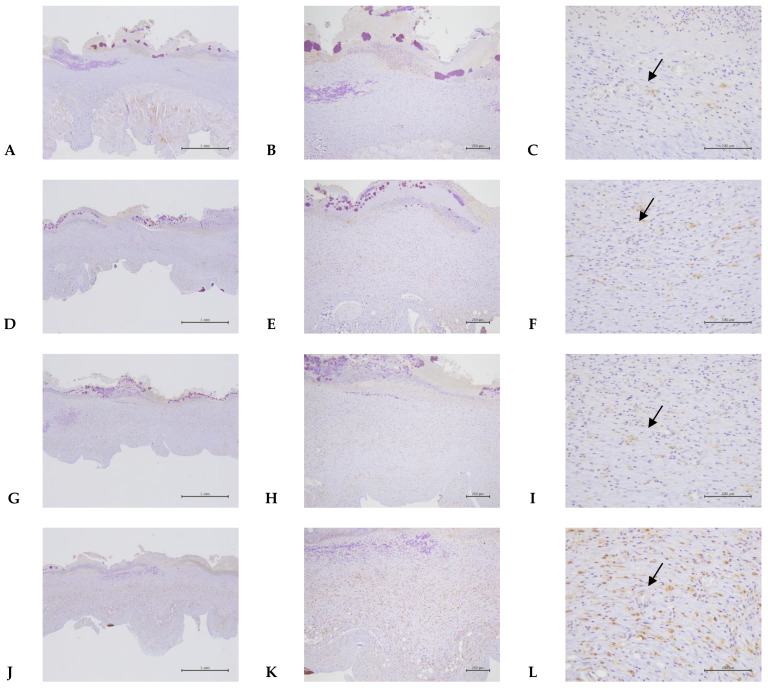
Immunohistochemistry staining of PEDOT: PSS (1.6) on wound healing in the full-thickness-excision wound model in normal rats on day seven (VEGF stain). Skin wounds displayed slight to moderate/severe angiogenesis (arrow) in G1: Control (**A**–**C**. Animal code: 1-2), G2 (**D**–**F**. Animal code: 2-5), G3 (**G**–**I**. Animal code: 3-2), and G4 (**J**–**L**. Animal code: 4-2). The VEGF staining frequency scores were 2.3 ± 0.4; 3.5 ± 0.5; 2.6 ± 1.5; and 4.2 ± 0.7 in G1, G2, G3, and G4, respectively. Magnification: 40×, 100×, and 400×.

**Figure 4 ijms-24-12539-f004:**
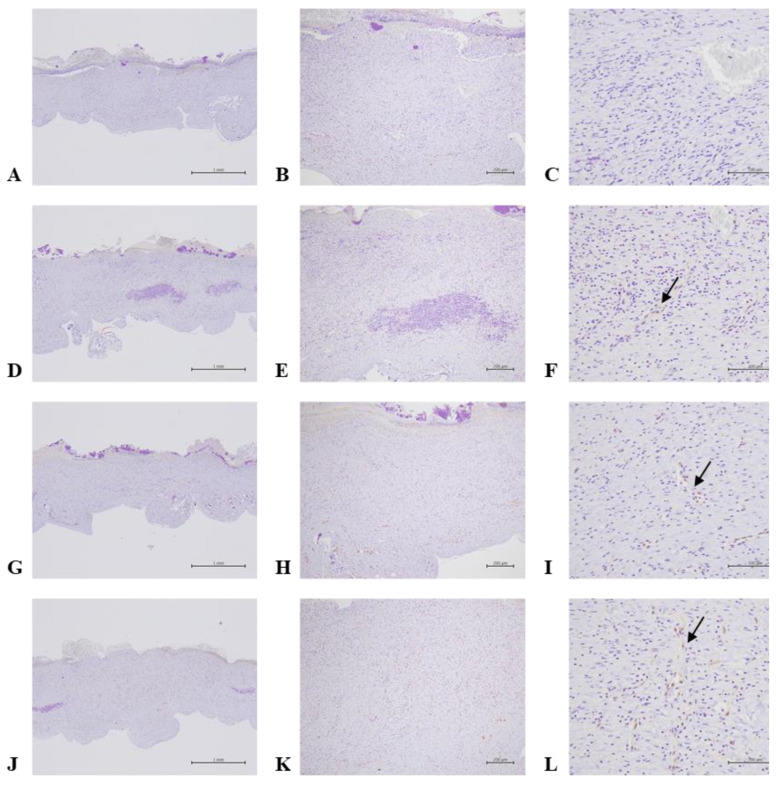
Immunohistochemistry staining of PEDOT: PSS (1.6) on wound healing in the full-thickness-excision wound model in normal rats on day 7 (TGF-β1 stain). Skin wounds showed an absence of TGF-β1 expression in the angiogenesis of blood vessels in G1: Control (**A**–**C**. Animal code: 1-3); however, minimal TGF-β1 expression in the angiogenesis of blood vessels (arrow) was noted in G2 (**D**–**F**. Animal code: 2-3) and G3 (**G**–**I**. Animal code: 3-2), and slight expression was noted in G4 (**J**–**L**. Animal code: 4-5). The TGF-β staining frequency scores were 0.0 ± 0.0; 0.3 ± 0.5; 0.2 ± 0.4; and 0.8 ± 0.7 in G1, G2, G3, and G4, respectively. Magnification: 40×, 100×, and 400×.

**Figure 5 ijms-24-12539-f005:**
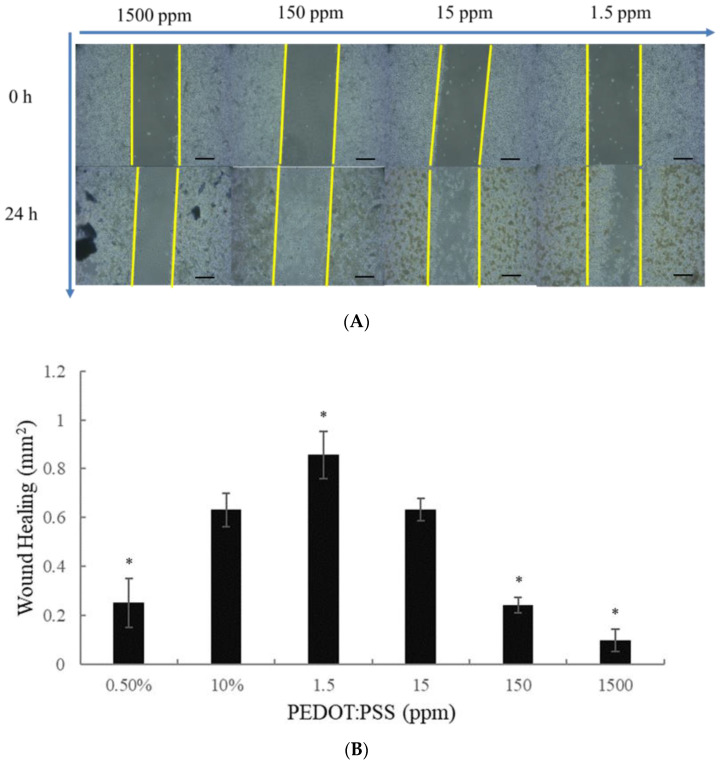
(**A**) PEDOT-PSS (1.6) 1.5 ppm markedly produced wound healing/closure in vitro. The experimental results showed significant wound healing after 24 h after wound healing in the macrophages (macrophage; RAW264.7) cells. The original magnification: 10×. And, the scale bar = 200 μM. (**B**) * *p* < 0.05 compared with the control group (according to one-way ANOVA).

**Figure 6 ijms-24-12539-f006:**
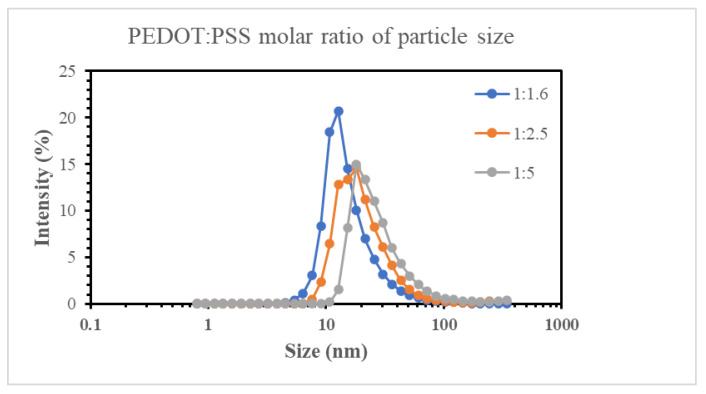
Different particle sizes between PEDOT: PSS (1.6), PEDOT: PSS (2.5), and PEDOT: PSS (5.0).

**Table 1 ijms-24-12539-t001:** PEDOT-PSS (1.6) 1.5 ppm showed an acceleration of the wound healing effects in animals.

Wound Area (mm^2^)					
Day Post Incision	0	3	7	11	14
Control	5142.2 ± 703.9	5045.2 ± 81.5	4928.1 ± 75.0	4187.2 ± 175.6	566.8 ± 69.5
PEDOT:PSS(1.6):1.5 ppm	5083.2 ± 315.2	4993.4 ± 107.2	3936.2 ± 136.0	2664.0 ± 211.4 *	349.5 ± 51.5 *

Wound photographs were taken on days 3, 7, 11, and 14 after the incision. * *p* < 0.05, compared to the control group (treated with normal control).

**Table 2 ijms-24-12539-t002:** Summary of pathological incidences of PEDOT: PSS on wound healing in the full-thickness-excision wound model in rats on days 7 and 14.

“Thickness Excision Wound”	Histopathological Lesions	Day 7			
		PEDOT: PSS			
		1	2	3	4
		Control	1.5 ppm	15 ppm	150 ppm
Skin					
	Crust, epidermis, slight to moderate/severe ^1^	5/5 ^3^	5/5	5/5	5/5
	Inflammation, slight to moderate/severe ^1^	5/5	5/5	5/5	5/5
Wound					
	Angiogenesis ^2^	5/5	5/5	5/5	5/5
	Granulation ^2^	5/5	5/5	5/5	5/5
	Re-epithelialization ^2^	0/5	0/5	0/5	0/5
“Thickness Excision Wound”	Histopathological lesions	Day 14			
		PEDOT: PSS			
		1	2	3	4
		Control	1.5 ppm	15 ppm	150 ppm
Skin					
	Crust, epidermis, slight to moderate/severe ^1^	2/4	1/5	1/5	2/5
	Inflammation, slight to moderate/severe ^1^	1/4	0/5	1/5	0/5
Wound					
	Angiogenesis ^2^	4/4	5/5	5/5	5/5
	Granulation ^2^	4/4	5/5	5/5	5/5
	Re-epithelialization ^2^	4/4	5/5	5/5	5/5

Group 1: control; groups 2~4: experimental groups. ^1^ The degree of lesions was graded from one to five depending on the severity: 1 = minimal (<1%); 2 = slight (1–25%); 3 = moderate (26–50%); 4 = moderate/severe (51–75%); 5 = severe/high (76–100%). ^2^ The degree of lesions was graded from one to four according to the method of Altavilla et al., 2001. ^3^ Incidence: affected skin/total skin was examined in rats (*n* = 4–5).

**Table 3 ijms-24-12539-t003:** Summary of VEGF staining intensity ^a^ and frequency ^b^ scores for angiogenesis markers of PEDOT: PSS (1.6) 1.5 ppm on wound healing in the full-thickness-excision wound model in rats on days 7 and 14.

Full-Thickness Excision Wound	Immunohistopathological Scores	Group			
		Day 7			
		1	2	3	4
		Control	PEDOT:PSS 1.5 ppm	PEDOT:PSS 15 ppm	PEDOT:PSS 150 ppm
VEGF					
	Staining intensity	1.3 ± 0.4	2.3 ± 0.4 *	2.7 ± 0.5 *	3.0 ± 0.0 *
	Staining frequency	2.3 ± 0.4	3.5 ± 0.5 *	3.7 ± 0.5 *	4.2 ± 0.7 *
Full-thickness excision wound	Histopathological lesions	Group			
		Day 14			
		1	2	3	4
		Control	PEDOT:PSS 1.5 ppm	PEDOT:PSS 15 ppm	PEDOT:PSS 150 ppm
VEGF					
	Staining intensity	2.0 ± 0.0	2.3 ± 0.4	2.2 ± 0.4	2.0 ± 0.0
	Staining frequency	3.0 ± 0.8	3.0 ± 0.0	2.4 ± 0.8	2.6 ± 0.5

^a^ The degree of staining intensity was graded from one to five depending on stained cells of a particular cell type or tissue element. 0: negative (no stained cells); 1 (±): equivocal (very faint stain); 2 (1+): weak (light stain); 3 (2+): moderate (light/medium stain); 4 (3+): strong (medium stain); 5 (4+): intense (dark stain). ^b^ The degree of staining frequency was graded from one to five depending on stained cells of a particular cell type or tissue element. 0: negative reaction (no stained cells); 1 (±) = minimal (< 5%); 2 (1+) = slight (5–25%); 3 (2+) = moderate (26–50%); 4 (3+) = moderate/severe (51–75%); 5 (4+) = severe/high (76–100%). * *p* < 0.05, compared to the control group (treated with normal control)

**Table 4 ijms-24-12539-t004:** Summary of TGF-β1 staining frequency ^a^ scores for angiogenesis markers of PEDOT: PSS (1.6) 1.5 ppm on wound healing in the full-thickness-excision wound model in rats on day 7.

Full-Thickness Excision Wound	Immunohistopathological Scores	Group			
		Day 7			
		1	2	3	4
		Control	PEDOT:PSS 1.5 ppm	PEDOT:PSS 15 ppm	PEDOT:PSS 150 ppm
TGF-β1	Angiogenesis				
	Staining frequency	0.0 ± 0.0	0.3 ± 0.5	0.2 ± 0.4	0.8 ± 0.7 *

^a^ The degree of staining frequency was graded from one to five depending on stained cells of a particular cell type or tissue element. 0: negative reaction (no stained cells); 1 (±) = minimal (< 5%); 2 (1+) = slight (5–25%); 3 (2+) = moderate (26–50%); 4 (3+) = moderate/severe (51–75%); 5 (4+) = severe/high (76–100%). * *p* < 0.05, compared to the control group (treated with normal control).

**Table 5 ijms-24-12539-t005:** Physical properties of various molar ratios of PEDOT:PSS.

Molar Ratio	1: 1.6	1:2.5	1:5.0
Conductivity (S/cm)	480	320	160

## Data Availability

At the moment, the data is not available due to privacy.

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
