# Peer review of "The Wound-Healing Activity of PEDOT-PSS in Animals"

_ijms, 2023, doi:10.3390/ijms241612539_

Round 1

Reviewer 1 Report

General remarks:
I suggest an extensive rephrasing of the manuscript, and ask for
editing help from someone with "full professional proficiency in English”.
the paragraph specific to statistical analyses is missing. I warmly suggest you add it.  

Introduction:
line 28 and 29, the sentences look missing a
connection

Line 44-45: “On the surface of the wound, after the skin is injured, the nearby tissue will gradually

proliferate microvascular, fibrous connective tissue to allow the wound to heal quickly.
the process is poorly described. Please expand.

Line 48: has a soft and soft appearance of granular appearance
I cannot understand what the authors are trying to communicate here.

Line 66: “As the PEDOT: PSS is similar in composition to graphene Can you expand a bit on the similarities please from the chemical point of view?

RESULTS:
PEDOT-PSS (1.6), PEDOT-PSS (2.5), and PEDOT-PSS (5.0)
is not really clear to me the differences between 1.6/2.5 and 5. I suggest reporting also here some details o refer to the materials section of the manuscript.

Figure 1: are 0.5% and 10% controls?
please change the color of the columns to distinguish them from the tested conditions, for example displaying them as white and the tests as black. In the caption, you mention only the 10%, what about the 0.5%? is 0.5% also serum?
is it a T-test or an ANOVA?
why did not you use the same amount for the 3 different PEDOT: PSS in the viability checking?
(you used similar but not identical amounts, from x-axis I can read 2.9; 2.5; 2.7)

Figure 2:
how did you compare the wounds? Please describe it in detail.

table 3: what is
organ?

PEDOT:

PSS15

0 ppm
please add space in front of 150 to not break the number in lines.

Line 163: “intensity and Frequency were slightly increased compared with the control group (Figure 4, Table 4).
how much is
slightly?
is possible for to you do a qPCR for TGF-beta and VEGF?
did you check some cytokines, for example IL1Beta for inflammation?

Materials: product code numbers are missing

Line 287: The hydrogel was designed for the wound healing assay in vivo can you describe the hydrogel?

Line 293: Different ratios result in different structures and variations in conductivity can you provide some references or test results?

Lines: 298 to 300 can you provide some references or test results?

Line 313-314: After 24 hours, pump out the solution in the well, add PBS to rinse, then added

PEDOT-PSS (1.6), and then put it at a constant temperature of 37°C for 30 minutes to react.
can you check this sentence? Perhaps here PEDOT-PSS is not the correct chemical you would write.

H&E Stain: lines 351-356 are unnecessary, please expand in place the technical details of your experiment.

I suggest an extensive rephrasing of the manuscript, and ask for editing help from someone with "full professional proficiency in English”.

Author Response

Dear professor: Thank you so much for evaluating this manuscript. Your assistance in this manuscript encourages us in exploring the study of PEDOT: PSS in the wound healing behavior. Importantly, the search for new substances or compositions that would help with such treatment is one of the current challenges. Your kind review and comments on this manuscript will stand in great esteem.

General remarks:

I suggest an extensive rephrasing of the manuscript, and ask for editing help from someone with "full professional proficiency in English”.

the paragraph specific to statistical analyses is missing. I warmly suggest you add it. 

Answer: thanks a lot for your review. This manuscript was sent to  https://www.mdpi.com/authors/english and the certificate was shown below (It will be shown in the file transferred.)

 As well, the statistical analyses section was added. Please see page 13, lines 398~405 in red pen.

Introduction:

line 28 and 29, the sentences look missing a connection

Answer: The sentence has been rewritten as “Although the wound healing process is not yet fully understood, it is known to involve the following tissue remodeling phases and their associated physiological actions: a vascular and inflammatory phase (hemostasis, inflammation), a tissue repair phase (fibroproliferation, cellular proliferation), and a tissue remodeling phase.”. Please see page 1, lines 26~30 in red pen.

Line 44-45: “On the surface of the wound, after the skin is injured, the nearby tissue will gradually proliferate microvascular, fibrous connective tissue to allow the wound to heal quickly.” the process is poorly described. Please expand.

Answer: I should make it clearer. The sentences have been rewritten as “These growth factors initiate the formation of granulation tissue, which gradually proliferates microvascular, fibrous connective tissue to allow the wound to heal quickly.”. Please see page 1, lines 41~42 in red pen.

Line 48: “has a soft and soft appearance of granular appearance”

I cannot understand what the authors are trying to communicate here.

Answer: I am sorry about that. The sentence should be written as “This gradual appearance of tissue covers an overall microvascular system, and the tis-sue will gradually turn bright red.”. Please see page 1, lines 43~44 in red pen.

Line 66: “As the PEDOT: PSS is similar in composition to graphene” Can you expand a bit on the similarities please from the chemical point of view?

Answer: Thanks for your suggestion. I understand that the physical properties for electronics between graphene and PEDOT: PSS is similar. However, the structure is not relevant. PEDOT: PSS is a "one-dimensional" linear polymer with conformationally mobile chains. This characteristic shows its better adhesion and penetration; however, graphene shows no conformational mobility. PEDOT: PSS presents itself with separated positive and negative charges that consists of linear flexible molecules seems preferable for biological applications. Importantly, PEDOT: PSS is devoid of toxicity and shows no carcinogens potential. Therefore, we decide to remove references to graphene altogether. Again, I appreciate your suggestion. Also, please see page 10, lines 217~222 in red pen.

RESULTS:

“PEDOT-PSS (1.6), PEDOT-PSS (2.5), and PEDOT-PSS (5.0)”

is not really clear to me the differences between 1.6/2.5 and 5. I suggest reporting also here some details refer to the materials section of the manuscript.

Answer: Please see the description of PEDOT: PSS in the Materials and Methods section “The polymer, PEDOT:PSS (1.6), PEDOT:PSS (2.5), and PEDOT:PSS (5.0) samples are aqueous dispersions, with the main components being poly(3,4-ethylenedioxythiophene) (PEDOT) and poly (styrenesulfonate) (PSS). The synthesis in-volves the polymerization of EDOT monomers with different ratios of PSS:PEDOT. The PSS: PEDOT molar ratios for PEDOT: PSS (1.6), PEDOT: PSS (2.5), and PEDOT: PSS (5.0) are 1:1.6, 1:2.5, and 1:5, respectively. Different ratios result in different structures and variations in conductivity, aiming to investigate cytotoxicity and compatibility issues. Moreover, to transform the solution-state PEDOT: PSS into a hydrogel for better adhesion during animal experiments, an aqueous thickening agent (provided by the First Cosmetics Works Limited) is added at a proportion of 1% of the total solution. This transformation helps effectively explore the impact of wound healing in animal stud-ies, especially with regard to easy application of hydrogen to the animals’ wounds.”. Please see page 11, lines 305~312 and page 12, lines 312~316 in red pen.

Figure 1: are 0.5% and 10% controls?

please change the color of the columns to distinguish them from the tested conditions, for example displaying them as white and the tests as black. In the caption, you mention only the 10%, what about the 0.5%? is 0.5% also serum?

Is it a T-test or an ANOVA? why did not you use the same amount for the 3 different PEDOT: PSS in the viability checking?

(you used similar but not identical amounts, from x-axis I can read 2.9; 2.5; 2.7)

Answer: Yes, thanks a lot for your suggestion. The color of the control and test groups has been changed and displayed them as white, and black, respectively. Yes, the 0.5% and 10% are the control groups. In our test, the 10% FBS did not show increase of the cell viability. But, we still presented these two concentration together. And, much sorry about for our typo. The correct of the concentrations tested should be expressed as the following: 2.9, 5.9, 11.7, 23.4, 46.9, 93.8, 187.5, 375, 750, and 1500 ppm. And, all the three different PEDOT: PSS should be all the same as mention-above. Also, the statistical analysis section has been added. This analysis was analyzed by one-way ANOVA. Please see page 3, lines 83~87 in a red pen.

Figure 2:

how did you compare the wounds? Please describe it in detail.

Answer: Wound areas of the animals significantly reduced at day 11 and day 14 after the full-thickness excision compared to the control group (Day 0) (Table 2). Please see page 4, lines 116~118 in a red pen.

table 3: what is “organ?”

Answer: I am so sorry about that. It should be expressed as “full-thickness excision wound”

Please see page 6, lines 141~143 in red pen.

PEDOT:PSS150 ppm

please add space in front of 150 to not break the number in lines.

Answer: thanks a lot for the suggestion. This has been corrected. Please see page 6, lines 141~143 in red pen.

Line 163: “intensity and Frequency were slightly increased compared with the control group (Figure 4, Table 4).”

how much is “slightly”? Is possible for to you do a qPCR for TGF-beta and VEGF?

did you check some cytokines, for example IL1Beta for inflammation?

Answer: Thanks for your valuable suggestion. The IHC test for TGF-β1 stain has been conducted at day 7. TGF-β1 may be correlated with the inflammation phase, which is happened at the beginning. Therefore, the TGF-β1 may not be detected at day 7. Therefore, our data only showed the significant effects of the TGF-β1 at higher concentration (analyzed by Student’s t-test). I appreciate your great suggestion that a qPCR for TGF-beta and VEGF should be conducted. And, check the IL1Beta? We absolutely will carry out this experiment next time in our further research.  Please see page 6, line 153 in red pen. and certain modification in page 7.

Materials: product code numbers are missing

Answer: Primary antibodies, anti-VEGF (product code number: ab32152), and anti-TGF-β1 (product code number: ab215715). Please see page 11, line 296~297 in red pen.

Line 287: “The hydrogel was designed for the wound healing assay in vivo” can you describe the hydrogel?

Answer: thanks a lot for the suggestions. The following shows the better description of the PEDOT: PSS hydrogel. The transformation from the aqueous to hydrogel form is beneficial for application of the medication in wound area. Their dosage applied in the wound area is the same as that used in aqueous form. Please see the followed description “Moreover, to transform the solution-state PEDOT: PSS into a hydrogel for better adhesion during animal experiments, an aqueous thickening agent (provided by the First Cosmetics Works Limited) is added at a proportion of 1% of the total solution. This transformation helps effectively explore the impact of wound healing in animal studies, especially with regard to easy application of hydrogen to the animals’ wounds.“ Please see page 11, line 311~312 and page 12, lines 313~316 in red pen.

Line 293: “Different ratios result in different structures and variations in conductivity” can you provide some references or test results?

Answer: Thanks a lot for the suggestion. We added the following experiment conducted by us earlier.

  1. The physical properties of PEDOT: PSS

The PSS:PEDOT molar ratios for PEDOT:PSS (1.6), PEDOT:PSS (2.5), and PEDOT:PSS (5.0) are 1.6, 1:2.5 , and 1:5.0, respectively. In the design of the PEDOT: PSS, the greater the amount of PSS, the poorer the conductivity of the material (Table 5). This is because PSS is non-conductive. Meanwhile, PEDOT: PSS shows the greatest conductivity. The reason that PEDOT: PSS is not available for our study is that PEDOT: PSS (1.0) is fairly unstable in wa-ter solution, and must be stored at a lower temperature. Our results also showed that PEDOT: PSS showed more intensity compared to PEDOT: PSS (2.5) and PEDOT: PSS (5.0) (Figure 6). Please see page 9, lines 201~215 in red pen.

Table 5: Physical properties of various molar ratio of PEDOT:PSS

Molar ratios

1: 1.6

1:2.5

1:5.0

Conductivity (S/cm)

480

320

160

Figure 6: Different particle sizes between PEDOT: PSS (1.6), PEDOT: PSS (2.5), and PEDOT: PSS (5.0).

Lines: 298 to 300 can you provide some references or test results?

Answer: We appreciate your valuable suggestion. The major approach to transform the solution into hydrogel is easily to apply the polymer in the animal wounds. We rewrite as the following paragraph. “The polymer, PEDOT: PSS (1.6), PEDOT: PSS (2.5), and PEDOT: PSS (5.0) samples are aqueous dispersions, with the main components being poly(3,4-ethylenedioxythiophene) (PEDOT) and poly (styrenesulfonate) (PSS). The synthesis in-volves the polymerization of EDOT monomers with different ratios of PSS:PEDOT. The PSS: PEDOT molar ratios for PEDOT: PSS (1.6), PEDOT: PSS (2.5), and PEDOT: PSS (5.0) are 1:1.6, 1:2.5, and 1:5, respectively. Different ratios result in different structures and variations in conductivity, aiming to investigate cytotoxicity and compatibility issues. Moreover, to transform the solution-state PEDOT: PSS into a hydrogel for better adhesion during animal experiments, an aqueous thickening agent (provided by the First Cosmetics Works Limited) is added at a proportion of 1% of the total solution. This transformation helps effectively explore the impact of wound healing in animal studies, especially with regard to easy application of hydrogen to the animals’ wounds.” Please see page 11, lines 305~314, and page 12, lines 315~317. We also presented certain characteristics of the materials in page 9, lines 201~215 in red pen.

Line 313-314: “After 24 hours, pump out the solution in the well, add PBS to rinse, then added

PEDOT-PSS (1.6), and then put it at a constant temperature of 37°C for 30 minutes to react.”

can you check this sentence? Perhaps here PEDOT-PSS is not the correct chemical you would write.

Answer: Thank you for the suggestion. We rewrite the sentence as following. “After that, PEDOT-PSS (1.6, 2.5, and 5.0) at various concentrations (2.9, 5.9, 11.7, 23.4, 46.9, 93.8, 187.5, 375, 750, and 1500 ppm) were added to each well, and three replicates were performed. After 24 hours, PBS was added for washing and a further 10 μL of the MTT was add into each well and incubated at 37 °C for 2h. Finally, the purple formazan crystals produced by the viable cells were measured at the absorbance length (590 nm) using an ELISA reader (Bio-Tek, Winooski, VT, USA).”. Please see page 12, lines 328~334 in red pen.

H&E Stain: lines 351-356 are unnecessary, please expand in place the technical details of your experiment.

Answer: Thanks a lot. Lines 351-356 has been deleted. The explanation of the experiment has been described as “The animal granulation tissue was removed and perfused with 4% paraformaldehyde for 30 minutes. The samples were fixed in 4% paraformaldehyde for 24 hours, embedded in paraffin, and cut into 7 mm cross-sections. The animal tissue sections were deparaffinized and rehydrated via gradient elution of ethanol and clearing in xylene, followed by staining with hematoxylin–eosin Y.  The slides were measured and photo-graphed using an Olympus BX53 fluorescence microscope (Tokyo, Japan). The staining intensity was analyzed using Image J software (National Institutes of Health, Bethesda, MD, USA). “. Please see page 13, lines 371~377 in red pen.

Comments on the Quality of English Language

I suggest an extensive rephrasing of the manuscript, and ask for editing help from someone with "full professional proficiency in English”.

Answer: Yes, we appreciate all what you have done for this paper. This manuscript has been corrected by MDPI English editing service.

Reviewer 2 Report

The treatment of hard-to-heal wounds is extremely relevant. The search for new substances or compositions that would help with such treatment is one of the current challenges.  The use of material from the field of electronics to treat wounds is original in itself. The authors wrote that this material is similar to graphene. I noted that from the point of view of physical properties for electronics - yes, it is. But from the point of view of structure it is absolutely wrong. Graphene is a two-dimensional "mesh" material. It has virtually no conformational mobility. The described polymer has different properties, indeed, it is a "one-dimensional" linear polymer with conformationally mobile chains. This conformational mobility gives it better adhesion and penetration, flexible is better than rigid in this case. Yes, graphene exhibits flexibility when adhering to a plane (in one dimension it has flexibility), but the environment of an organism is more complex than a plane. Therefore, a polymer with separated positive and negative charges that consists of linear flexible molecules seems preferable for biological applications. Additionally, I would point out that graphene can roughly be considered a polyaromatic hydrocarbon, which are generally recognized as carcinogens. The described polymer is devoid of this disadvantage.

Only the claims about graphene need to be corrected. Perhaps we should remove references to graphene altogether. The authors are testing a new material for wound healing and it shows the expected properties well.

Line 66. About graphene. The ideal graphene is carbon. Taking into account the edge effect, we can call graphene a hydrocarbon. The polymer used is rich in functional groups and its is only similar to graphene in terms of physical properties but not in composition.

Author Response

Dear professor: Thank you so much for evaluate this manuscript. Your assistance in this manuscript encourages us in exploring the effects of PEDOT: PSS in the wound healing study. Importantly, the search for new substances or compositions that would help with the clinician to treat the wound healing, especially, further study will be focused on the chronic wound healing is one of the current challenges. Your kind review and comments on this manuscript will stand in great esteem.

Comments and Suggestions for Authors

The treatment of hard-to-heal wounds is extremely relevant. The search for new substances or compositions that would help with such treatment is one of the current challenges.  The use of material from the field of electronics to treat wounds is original in itself. The authors wrote that this material is similar to graphene. I noted that from the point of view of physical properties for electronics - yes, it is. But from the point of view of structure it is absolutely wrong. Graphene is a two-dimensional "mesh" material. It has virtually no conformational mobility. The described polymer has different properties, indeed, it is a "one-dimensional" linear polymer with conformationally mobile chains. This conformational mobility gives it better adhesion and penetration, flexible is better than rigid in this case. Yes, graphene exhibits flexibility when adhering to a plane (in one dimension it has flexibility), but the environment of an organism is more complex than a plane. Therefore, a polymer with separated positive and negative charges that consists of linear flexible molecules seems preferable for biological applications. Additionally, I would point out that graphene can roughly be considered a polyaromatic hydrocarbon, which are generally recognized as carcinogens. The described polymer is devoid of this disadvantage.

Only the claims about graphene need to be corrected. Perhaps we should remove references to graphene altogether. The authors are testing a new material for wound healing and it shows the expected properties well.

Answer: Thanks a lot for your valuable information. I definitely will take the suggestion and put in a approximate place of this manuscript. I really appreciate. And, all of the description of graphene were removed.

Line 66. About graphene. The ideal graphene is carbon. Taking into account the edge effect, we can call graphene a hydrocarbon. The polymer used is rich in functional groups and its is only similar to graphene in terms of physical properties but not in composition.

Answer: Yes, thanks again for sharing me this valuable information. All of the information of graphene were removed from this manuscript. Our results from the viability assay of PEDOT: PSS (1.6) actually showed no toxicity even the concentration goes up to 1500 ppm. Therefore, this polymer (PEDOT: PSS) may have its potential biological activity. We are currently conducting the toxicity test (28 days, 90 days, and genetic toxicity tests) for PEDOT: PSS. Until the results were collected, the manuscript will be submitted to the paper in the near future.

Round 2

Reviewer 1 Report

authors answered all my questions